# Comparative Assessment of Risk and Turn-Around Time between Sequence-Based Typing and Next-Generation Sequencing for HLA Typing

**DOI:** 10.3390/diagnostics14161793

**Published:** 2024-08-16

**Authors:** Jaehyun Cha, Mina Hur, Hanah Kim, Seunggyu Yun, Myunghyun Nam, Yunjung Cho, Minjeong Nam

**Affiliations:** 1Department of Laboratory Medicine, Korea University Anam Hospital, Seoul 02841, Republic of Korea; chajh0901@gmail.com (J.C.); koryun@korea.ac.kr (S.Y.); yuret@korea.ac.kr (M.N.); eqcho1ku@korea.ac.kr (Y.C.); 2Department of Laboratory Medicine, Konkuk University School of Medicine, Seoul 05030, Republic of Korea; dearmina@hanmail.net (M.H.);; 3Department of Laboratory Medicine, Korea University College of Medicine, Seoul 02841, Republic of Korea

**Keywords:** sequence-based typing, next-generation sequencing, risk, turn-around time, failure mode and effect analysis

## Abstract

This study compared laboratory risk and turn-around time (TAT) between sequence-based typing (SBT) and next-generation sequencing (NGS) for human leukocyte antigen (HLA) typing. For risk assessment, we utilized the risk priority number (RPN) score based on failure mode and effect analysis (FMEA) and a risk acceptability matrix (RAM) according to the Clinical Laboratory Standards Institute (CLSI) guidelines (EP23-A). Total TAT was documented for the analytical phase, and hands-on time was defined as manual processes conducted by medical technicians. NGS showed a significantly higher total RPN score than SBT (1169 vs. 465). NGS indicated a higher mean RPN score, indicating elevated severity and detectability scores in comparison to SBT (RPN 23 vs. 12, *p* = 0.001; severity 5 vs. 3, *p* = 0.005; detectability 5 vs. 4, *p* < 0.001, respectively). NGS required a greater number of steps than SBT (44 vs. 25 steps), all of which were acceptable for the RAM. NGS showed a longer total TAT, total hands-on time, and hands-on time per step than SBT (26:47:20 vs. 12:32:06, 03:59:35 vs. 00:47:39, 00:05:13 vs. 00:01:54 hh:mm:ss, respectively). Transitioning from SBT to NGS for HLA typing involves increased risk and an extended TAT. This study underscored the importance of evaluating these factors to optimize laboratory efficiency in HLA typing.

## 1. Introduction

Human leukocyte antigen (HLA) typing plays a pivotal role in determining compatibility between donor and recipients in transplantation and providing insights into the genetic basis of autoimmune diseases [1,2,3,4]. HLA typing can be classified into three resolution levels: low resolution (serological-level typing), medium resolution (allelic group typing), and high resolution (allelic-level typing) [1]. Serological-level typing indicated lower sensitivity and specificity, along with reported inconsistencies, prompting the introduction of DNA-based testing [5]. DNA-based testing is considered a reliable alternative, identifying two alleles of major HLA loci—HLA-A, -B, -C, -DRB1, -DQB1, and -DPB1 [6].

Sequence-based typing (SBT) is considered the gold standard at the allele level for determining the full nucleotide sequence of the alleles present [7]. However, SBT can result in genotype ambiguity, allele ambiguity, and phase ambiguity. A growing list of ambiguities required additional testing, leading to a time-consuming process, additional costs, and delayed results [7]. Next-generation sequencing (NGS) has recently been actively introduced for HLA typing, utilizing the massive parallel sequencing of clonal DNA molecules [8,9]. This approach significantly increased genomic coverage and effectively improved ambiguity resolution [10].

Clinical laboratories are consistently under pressure to provide qualified and reliable results within a limited time, without compromising laboratory efficiency [11]. Risk assessment contributes to enhancing patient safety and satisfaction, ensuring the delivery of high-quality laboratory results [12]. Laboratory efficiency can be achieved by decreased risk and turn-around time (TAT) [13]. A shorter TAT facilitates early diagnosis and treatment, reduces hospital stays, and improves patient satisfaction [14,15].

According to the International Organization for Standardization (ISO) 22367:2020 and Clinical Laboratory Standards Institute (CLSI) guidelines (EP18-A2), it was recommended to conduct a failure mode and effect analysis (FMEA) as part of the risk assessment when introducing new test methods or equipment into a clinical laboratory [16,17]. FMEA systematically identifies and evaluates potential failures and their causes across the entire process by a top-down approach [17]. The risk associated with each step was determined by calculating a risk priority number (RPN) score, derived from the multiplication of severity, occurrence, and detectability scores [18]. In addition, ISO 14971:2007 and CLSI guidelines EP23-A have introduced a risk acceptability matrix (RAM) for qualitative risk evaluation [19,20]. This matrix determined the risk based on the combination of two factors: the severity and occurrence of risk.

Various fields, such as clinical chemistry, diagnostic hematology, and transfusion medicine, have actively assessed the risks associated with testing methods to improve their laboratory efficiency [21,22]. However, to our knowledge, there have been no studies evaluating FMEA and RAM for HLA typing methods. Thus, this study aimed to evaluate the relative risk of SBT, considered the gold standard for high-resolution HLA typing compared to the newly introduced NGS, using FMEA and RAM. A comparative analysis of the TAT was also conducted to gain insights into the multifaceted aspects of SBT and NGS.

## 2. Materials and Methods

### 2.1. Sample Collection

This comparative study was conducted between July 2022 and May 2023 at Korea University Medical Center (KUMC) in Seoul, South Korea. The Institutional Review Board of KUMC exempted approval for the study protocol (K2023-0107) and waived the requirement for informed consent, given the utilization of the remaining samples after testing.

### 2.2. HLA Typing Using SBT and NGS

We analyzed samples for HLA-A, -B, -DRB1, and -DQB1 typing, with genomic DNA extracted from peripheral blood collected during the study. The initial steps, from sample collection to DNA extraction, were excluded from the comparative assessment as the process was identical for both SBT and NGS. The evaluation of risk and TAT for SBT and NGS was conducted from DNA amplification to the reporting of results. SBT was performed using AVITA plus^TM^ HLA SBT kits (Biowithus Inc., Seoul, Republic of Korea) and BigDye Terminator mixture (Biowithus Inc.) according to the manufacturer’s instructions. Polymerase chain reaction (PCR) amplification, PCR product clean-up, and sequencing reaction were performed using GeneAmp^®^ PCR System 9700 Thermal Cycler (PE-Biosystems, Foster City, CA, USA). The sequenced results were analyzed using BIOWITHUS SBT Analyzer software ver. 2.7.5 (Biowithus Inc.). NGS was performed using NGSgo^®^-AmpX c2 HLA kit (GenDx, Utrecht, The Netherlands) and NGSgo^®^ Library Full kit (GenDx, Utrecht, Holland) according to the manufacturer’s instructions. PCR amplification, fragmentation with adaptor ligation, and indexing PCR were performed using GeneAmp^®^ PCR System 9700 Thermal Cycler (PE-Biosystems, Foster City, CA, USA). Library amplification and sequencing were performed on a Miseq platform (Illumina Inc., San Diego, CA, USA). Sample genotypes were assigned using NGSengine version 2.30.0.28772 from GenDx based on the IPD-IMGT/HLA Database version 3.51.0.

### 2.3. Assessment of Risk and TAT

Risk assessment followed a systematic six-step process, as follows: (1) determination of the analytical test, (2) assembly of a team, (3) risk identification, (4) risk analysis, (5) risk assessment, and (6) action for risk improvement. The risk assessment team comprised a director, a medical doctor, and three qualified medical technicians. The director, experienced with FMEA, trained the other team members to identify potential defects, preventive actions, and outcomes. Team members independently conducted their risk assessment by reviewing each step of SBT or NGS and identifying risks based on their experience and information in the literature. Any discrepancies were analyzed and discussed until a consensus was reached for objectivity. For the risk assessment and evaluation, two tools, FMEA and RAM, were used. Using the FMEA rating scale, severity, occurrence, and detectability scores were assigned for each step [16,17,23]. Each score was determined as the mean of the scores assigned by the team members. The overall RPN score was calculated by multiplying the severity, occurrence, and detectability scores. Using the RAM rating scale, severity and occurrence were assigned for each step. Severity was categorized into five levels: negligible (inconvenience or temporary discomfort); minor (temporary injury or impairment not requiring medical intervention); serious (injury or impairment requiring medical intervention); critical (permanent impairment or life-threatening injury); and catastrophic (patient death). Occurrence was categorized into five levels: frequent (once per week); probable (once per month); occasional (once per year); remote (once every few years); and improbable (once during system measurement). Each step was judged as acceptable or unacceptable according to the classification described in Table 1. After the risk assessment and evaluation for each step, team members suggested preventive actions for steps with high RPN scores or unacceptable RAM results.

The TATs for each step in both SBT and NGS were recorded using a digital camera and stopwatch. The recording process was repeated three times to ensure precision in the comparison. The mean value of the three measurements was determined as the TAT for each step. The total TAT was calculated as the sum of TATs for all steps, further categorized into hands-on time and machine running time. Hands-on time represented manual steps performed by medical technicians during the test, while machine running time represented the non-manual steps.

### 2.4. Statistical Analysis

A Mann–Whitney U test was utilized for the nonparametric unpaired comparison between SBT and NGS. Statistical analysis was conducted using IBM SPSS v.26 (IBM Corp., Armonk, NY, USA), MedCalc v.19.1.7 (MedCalc Software, Ostend, Belgium), and Microsoft Excel 2013 on Windows 10. *p*-values < 0.05 were considered indicative of statistical significance.

## 3. Results

In SBT, the most common action for a potential defect was a repeat of the step, with outcomes resulting in delays or incorrect results (Table 2). According to the FMEA rating scale, the severity score ranged from 2 to 8, occurrence score ranged from 1 to 2, and detectability score ranged from 1 to 6. The resulting RPN scores for each step ranged from 3 to 72. Based on the RAM rating scale, severity showed 1 minor and 24 negligible steps, and occurrence showed 1 occasional, 5 remote, and 19 improbable steps. All steps revealed varying levels of risk, but according to the RAM rating scale, the risks were clinically acceptable.

Similarly, in NGS, a repeat of the step and delay or incorrect results were the most common actions and outcomes (Table 3). According to the FMEA rating scale, the severity score ranged from 3 to 8, occurrence score ranged from 1 to 2, and detectability score ranged from 3 to 6. The resulting RPN scores for each step ranged from 12 to 102. Based on the RAM rating scale, severity showed 1 minor and 43 negligible steps, and occurrence showed 1 occasional, 11 remote, and 32 improbable steps. All steps revealed varying levels of risk, but according to the RAM rating scale, the risks were clinically acceptable.

In SBT, the shortest step for hands-on time and overall TAT were the PCR product clean-up step and the interpretation and report step (00:02:30, 00:11:28 hh:mm:ss, respectively) (Table 4). The longest hands-on time, machine running time, and overall TAT were purification for the dye removal step (00:19:32, 05:56:57, 06:16:29 hh:mm:ss, respectively).

In NGS, the shortest hands-on time and overall TAT were the library quantification step (both 00:09:04 hh:mm:ss) (Table 5). The longest hands-on time was the interpretation and reporting step, (01:01:40 hh:mm:ss) and the longest machine running time and overall TAT were the next-generation sequencing step (17:00:00 and 17:13:43 hh:mm:ss, respectively).

NGS had a higher number of total steps than SBT (44 vs. 25 steps) (Table 6). The risks at all steps in both SBT and NGS were clinically acceptable. NGS showed a higher total RPN score than SBT (1169 vs. 465 RPN). NGS showed a higher mean RPN score with higher mean severity and detectability scores than SBT (RPN 23 vs. 12, *p* = 0.001; S 5 vs. 3, *p* = 0.005; D 5 vs. 4, *p* < 0.001, respectively). NGS indicated longer total TAT, total hands-on time, and hands-on time/step than SBT (26:47:20 vs. 12:32:06, 03:59:35 vs. 00:47:39, 00:05:13 vs. 00:01:54 hh:mm:ss, respectively).

## 4. Discussion

We conducted a comparative analysis of risk and TAT between SBT and NGS for HLA typing. Our findings revealed that the total RPN score of NGS was higher than SBT (1169 vs. 465 RPN), primarily due to the increased number of processing steps (44 vs. 25 steps). The median RPN score in NGS was also higher than SBT (23 vs. 12 RPN). NGS showed higher median severity and detectability scores than SBT (severity 5 vs. 3; detectability 5 vs. 4). Despite the elevated risk of NGS compared to SBT, it remained acceptable according to the RAM rating scale. The total TAT, total hands-on time, and hands-on time/step of NGS were longer than SBT (26:47:20 vs. 12:32:06, 03:59:35 vs. 00:47:39, 00:05:13 vs. 00:01:54 hh:mm:ss, respectively).

Recently, there has been a growing interest in assessing laboratory efficiency in clinical laboratories to improve productivity and patient safety [11]. The assessments involved the evaluation of risk assessment, TAT, and costs. FMEA, initially used in the industrial field, has recently found application in laboratory risk assessment. In a previous study by Nam et al. [24], a risk assessment comparison using FMEA was conducted between the tube test and automated column agglutination technology for anti-A/B isoagglutinin titers. The automated column agglutination technology demonstrated lower RPN scores than the tube test (337 vs. 1843 RPN), with the manual serial dilution step of the tube test accompanying the highest RPN score of 1080. This finding highlighted that RPN scores significantly increased when a processing step involved manual procedures, required skilled techniques, or included repetitive and numerous steps. NGS involved numerous complex steps, such as DNA fragmentation, adapter ligation, library pooling, size selection, and library quantification. The critical steps in NGS required more manual procedures and skilled techniques compared to SBT, leading to increased mean severity, mean detectability, mean RPN, and total RPN score. RAM, another indicator of laboratory risk management suggested by CLSI EP23-A, was also evaluated in this study [19]. Although the risk of NGS was higher than SBT, NGS was still deemed acceptable for HLA typing. Since there is currently no research applying the RAM in laboratory efficiency, further research is needed.

Our novel findings involved a comparison between FMEA and RAM. FMEA demonstrated significant advantages over RAM in comparing two or more items by quantifying the risk through scores [17]. However, a disadvantage of FMEA was its inability to establish a clear standard for evaluating the true acceptability of each processing step [23]. There was no standardized or validated cutoff for RPN scores. Nevertheless, Han et al. [23] suggested a careful review of the steps exceeding a score of 300 RPN. In this study, none of the processing steps in SBT and NGS exceeded 300 RPN. Regarding RAM, the rating scale appeared to be a more suitable method than FMEA for performing conformity assessments at each step [19]. However, RAM was not suitable for performing comparisons between different methods. In this study, a risk comparison between SBT and NGS could not be performed using RAM due to the same proportion of acceptable steps. Each laboratory should consider the appropriate criteria for conducting a risk assessment or evaluation based on its purpose.

TAT, another critical indicator of laboratory efficiency, has been actively used in previous research. Kim et al. [25] measured the mean, 99th percentile, and CV% TAT to assess the laboratory efficiency of total laboratory automation. The mean TAT, representing the timeliness of result reporting, decreased by 6.1%; the 99th percentile TAT, representing the outlier rate, decreased by 13.3%; and the CV% TAT, representing predictability, decreased by 70.0%. Despite the effectiveness of TAT in assessing laboratory efficiency, there is no clear consensus on which period should be incorporated when establishing TAT for a specific test [26,27]. In this study, we did not analyze the pre-analytical phase and DNA extraction step due to performing the same process in SBT and NGS. TATs were assessed after DNA extraction until the final reporting. Similar to risk assessment, TAT increased when the test involved manual procedures, required skilled techniques, or included repetitive and numerous steps. Due to the complex and numerous steps involved in NGS, NGS indicated longer total TAT, total hands-on time, and particularly hands-on time/step.

According to several reports, an increase in hands-on times has been correlated with increasing medical costs. Shin et al. [28] reported labor costs into the total expense, considering the average technician salary and the hands-on time required for testing a single sample. In the case of unexpected antibody screening, the automated analyzer significantly reduced labor costs compared to manual methods (KRW 539 vs. 2961 Korean), subsequently reducing the total expense (KRW 5920 vs. 7157 Korean). In this study, the increased hands-on time in NGS could potentially elevate labor costs, consequently contributing to an overall increase in total expense. However, we did not comprehensively compare costs in this study; further research is needed to compare direct and indirect costs between NGS and SBT.

Recent studies have shown controversial results regarding the impact of laboratory automation in terms of laboratory efficiency. Chung et al. [21] revealed that the automated crossmatching method has a lower RPN score (229 vs. 1435 RPN), shorter TAT (19.1 vs. 23.3 min), shorter hands-on time (1.1 vs. 5.3 min), and lower costs/test (USD 1.44 vs. 2.70) than the manual crossmatching method. Nam et al. [24] found that automated column agglutination technology showed lower RPN scores (33,700 vs. 184,300 RPN) than manual tube tests for anti-A/B isoagglutinin titers. However, TAT and cost were similar in automated and manual methods (TAT, 15:23:00 vs. 14:26:40; cost, 1377.4 vs. 1312.4, respectively). Nam et al. [22] revealed that automated white blood cell counting in leukopenic samples showed an even longer TAT than manual counting. In this study, NGS showed an increased risk and TAT due to the increase in the number and complexity of steps. Despite the controversial results on laboratory efficiency for automation, we expect that the automation of several NGS processes can improve laboratory efficiency. In particular, the DNA clean-up step revealed the highest RPN score, and the DNA quantification step revealed a remarkably longer hands-on time. The automation of these steps can contribute to improving the laboratory efficiency of NGS.

This study has some limitations. It involved a comparison of risk and TAT for a single sample without the evaluation of multiple samples. Given that NGS utilized a flow cell for analyzing multiple samples simultaneously, there is potential for a significant reduction in risk and TAT. In addition, TAT might differ between qualified and unskilled technicians. Therefore, it will be necessary to minimize the possibility of variation by conducting multiple measurements of the performance of a qualified and skilled technician.

## 5. Conclusions

This is the first study to compare risk and TAT between NGS and SBT for HLA typing. NGS required more processing steps and a more significant manual workload than SBT, resulting in higher RPN scores and longer TATs. These results suggest that it is necessary to consider the high risk and long TAT in NGS when contemplating the transition from SBT to NGS for HLA typing. Furthermore, companies involved in NGS development need to optimize the testing workflow, reduce the manual workload, and increase the automation process.

## Figures and Tables

**Table 1 diagnostics-14-01793-t001:** Rating scale of severity and occurrence by using * risk acceptability matrix.

		Severity				
		Negligible	Minor	Serious	Critical	Catastrophic
Occurrence	Frequent	U	U	U	U	U
	Probable	A	U	U	U	U
	Occasional	A	A	A	U	U
	Remote	A	A	A	A	U
	Improbable	A	A	A	A	A

* Risk acceptability matrixsourced from CLSI EP23-A [19].

**Table 2 diagnostics-14-01793-t002:** Risk assessment in SBT for HLA typing.

Processing Step	Potential Defect	Action	Outcome	FMEA				RAM	
				S	O	D	RPN	S	O
Amplification									
1. Add PCR mixture to PCR tubes	IA, MA	R	D, IR	4	1	5	20	N	I
2. Add extracted DNA to PCR tubes	IA, MA	R	D, IR	8	1	5	44	M	I
3. Run in PCR system	Incorrect conditions	R	D, IR	4	1	3	10	N	I
Gel electrophoresis									
4. Add PCR product to agarose gel	IA, MA	R	D, IR	2	1	4	12	N	Rm
5. Electrophoresis	Incorrect conditions	R	D, IR	3	1	2	5	N	I
6. Interpret the bands with an analyzer	Incorrect interpretation	R	IR	3	1	2	5	N	I
PCR product clean-up									
7. Add clean solution and run in the PCR system	Incomplete cleaning	R	D, IR	4	1	4	18	N	I
Sequencing reaction									
8. Dilute the PCR products	Incorrect conditions	R	D, IR	4	1	5	26	N	Rm
9. Add primer and dye mixture	IA, MA	R	D, IR	6	2	5	48	N	Rm
10. Add diluted PCR products	IA, MA	R	D, IR	7	2	5	72	N	Oc
11. Run in PCR system	Incorrect conditions	R	D, IR	4	1	2	10	N	I
Purification for dye removal									
12. Add sodium acetate/EDTA buffer	IA, MA	R	D, IR	3	1	5	21	N	Rm
13. Add ethanol to the plate	IA, MA	R	D, IR	3	1	5	13	N	I
14. Centrifuge (30 min at 2000× *g*)	Spill, mechanical error	R	D	3	1	1	3	N	I
15. Remove supernatant	Insufficient removal	R	D	2	1	5	11	N	I
16. Add 80% ethanol	IA, MA	R	D, IR	3	1	4	12	N	I
17. Centrifuge (5 min at 2000× *g*)	Spill, mechanical error	R	D	3	1	1	3	N	I
18. Remove supernatant	Insufficient removal	R	D	2	1	5	11	N	I
19. Dry 3 min at 65 °C	Incorrect conditions	R	D	2	1	2	5	N	I
20. Add high-deionized formamide	IA, MA	R	D, IR	4	1	4	16	N	I
21. Dry 3 min at 95 °C	Incorrect conditions	R	D	2	1	2	4	N	I
22. Incubate for 3 min at 0 °C	Incorrect conditions	R	D, IR	2	1	2	4	N	I
23. Run in genetic analyzer	Insufficient volume	Retest	D	5	1	3	14	N	I
Interpretation and report									
24. Interpret the results	Incorrect interpretation	R	IR	6	1	6	45	N	Rm
25. Input the result on LIS manually	Clerical error	Correct	IR	5	1	6	33	N	I
Total							465		

Abbreviations: SBT, sequence-based typing; HLA, human leukocyte antigen; FMEA, failure mode and effect analysis; RAM, risk acceptability matrix; S, severity; O, occurrence; D, detectability; RPN, risk priority number; PCR, polymerase chain reaction; IA, incorrect amounts; MA, missed addition; R, repeat; D, delay; IR, incorrect result; N, negligible; I, improbable; DNA, deoxyribonucleic acid; M, minor; temp., temperature; Rm, remote; Oc, occasion; EDTA, ethylenediamine tetraacetic acid; LIS, laboratory information system.

**Table 3 diagnostics-14-01793-t003:** Risk assessment and evaluation in NGS for HLA typing.

Processing Step	Potential Defect	Action	Outcome	FMEA				RAM	
				S	O	D	RPN	S	O
Amplification									
1. Add nuclease-free water to PCR tube	IA, MA	R	D, IR	4	1	4	15	N	I
2. Add polymerase, buffer, and dNTPs	IA, MA	R	D, IR	5	1	5	23	N	I
3. Identify and add primer pellet	IA, MA	R	D, IR	5	2	5	44	N	Rm
4. Add extracted DNA to reaction mix	IA, MA	R	D, IR	8	1	5	42	M	I
5. Run in PCR system	Incorrect conditions	R	D, IR	5	1	3	13	N	I
DNA quantification									
6. Add Qubit reagent to Qubit tube	IA, MA	R	D, IR	4	1	5	18	N	I
7. Add Qubit buffer to Qubit tube	IA, MA	R	D, IR	4	1	5	18	N	I
8. Add DNA to Qubit tube	IA, MA	R	D, IR	3	1	5	17	N	I
9. Measure amplicons conc. at fluorometer	Incorrect interpretation	R	IR	4	2	3	20	N	Rm
10. Calculate the required volume of each gene and pool amplicons	Incorrect calculation	R	IR	4	2	5	36	N	Rm
Fragmentation and adapter ligation									
11. Prepare fragmentation master mix	Contamination	R	D	4	1	5	20	N	I
12. Add master mix to plate well	IA, MA	R	D, IR	5	1	5	25	N	I
13. Add amplicon to plate well	IA, MA	R	D, IR	5	1	5	27	N	I
14. Incubation in a PCR system at 25 °C for 20 min and 70 °C for 10 min	Incorrect conditions	R	D, IR	4	1	3	12	N	I
15. Prepare adaptor ligation master mix	IA, MA	R	D, IR	5	1	5	27	N	I
16. Add ligation master mix to DNA fragmentation	IA, MA	R	D, IR	5	1	5	25	N	I
17. Incubation in PCR system at 20 °C for 15 min	Incorrect conditions	R	D, IR	5	1	3	13	N	I
DNA clean-up									
18. Add magnetic beads and incubate for 5 min	IA, MA	R	D, IR	4	1	5	22	N	I
19. Place the plate on a magnetic stand and remove supernatant	Insufficient removal	R	D	4	2	3	16	N	Rm
20. * Add 80% ethanol, incubate for 30 s, and remove the supernatant	IA, MA	R	D, IR	4	2	5	102	N	Rm
21. Remove ethanol and dry 3~5 min	Excessive drying, missed removal	R	D, IR	4	2	5	40	N	Oc
22. Add elution buffer and incubate for 2 min at RT on shaker	IA, MA	R	D, IR	4	1	3	13	N	I
23. Place the plate on a magnetic stand and transfer eluate to a new plate	IA, contamination	R	D	4	1	5	20	N	I
Indexing PCR									
24. Prepare (thaw and centrifuge) for reagent	Insufficient thawing, contamination	R	D	4	1	5	20	N	I
25. Add HiFi PCR Mix to IndX plate	IA, MA	R	D, IR	5	1	5	27	N	I
26. Add eluate to IndX plate	IA, MA	R	D, IR	5	1	5	28	N	I
27. Run in PCR system	Incorrect conditions	R	D, IR	5	1	3	13	N	I
Library pooling, DNA clean-up, and size selection									
28. Transfer DNA to new tube and mix	IA, contamination	R	D	5	1	5	25	N	I
29. Add magnetic beads and incubate for 5 min	IA, MA	R	D, IR	5	1	5	23	N	I
30. Place the plate on a magnetic stand and remove the supernatant	Insufficient removal	R	D	4	2	5	31	N	Rm
31. Add 80% ethanol, incubate for 30 s, and remove supernatant	IA, MA	R	D, IR	5	2	5	36	N	Rm
32. Remove ethanol and dry 3~5 min	Excessive drying, missed removal	R	D, IR	4	2	5	31	N	Rm
33. Add elution buffer and incubate for 2 min at RT on shaker	IA, MA	R	D, IR	5	1	3	14	N	I
34. Place the plate on a magnetic stand and transfer eluate to a new tube	IA, contamination	R	D	4	1	5	21	N	I
Library quantification									
35. Prepare Qubit working solution with library sample	IA, MA	R	D, IR	5	1	5	23	N	I
36. Measure library conc. at a fluorometer and calculate Qubit results to nanomolar conc.	IA, MA	R	D, IR	5	2	3	26	N	Rm
Next-generation sequencing									
37. Thaw and wash for reagent	Contamination	R	D	4	1	5	20	N	I
38. Dilute library to a conc. of 4 nM	Incorrect conditions	R	D, IR	6	2	5	50	N	Rm
39. Add 0.2N NaOH	IA, MA	R	D, IR	5	1	5	23	N	I
40. Incubation the tube	Incorrect conditions	R	D, IR	3	1	5	17	N	I
41. Add HT1 buffer	IA, MA	R	D, IR	5	1	5	23	N	I
42. Run paired-end sequencing in Miseq analyzer	Insufficient volume	Retest	D	5	1	3	12	N	I
Interpretation and report									
43. Interpret the results	Incorrect interpretation	R	IR	7	1	6	62	N	Rm
44. Input the result on LIS manually	Clerical error	Correct	IR	6	1	6	36	N	I
Total							1169		

* Step 20 was repeated 3 times and the RPN score was multiplied by 3. Abbreviations: NGS, next-generation sequencing; HLA, human leukocyte antigen; FMEA, failure mode and effect analysis; RAM, risk acceptability matrix; S, severity; O, occurrence; D, detectability; RPN, risk priority number; PCR, polymerase chain reaction; IA, incorrect amounts; MA, missed addition; R, repeat; D, delay; IR, incorrect result; N, negligible; I, improbable; dNTPs, deoxynucleotide triphosphates; Rm, Remote; M, minor; temp., temperature; conc., concentration; Oc, occasion; RT, room temperature; HT, hybridization; LIS, laboratory information system.

**Table 4 diagnostics-14-01793-t004:** TAT assessment and evaluation in SBT for HLA typing.

Processing Step	TAT	Hands-On Time	Machine Running Time	Overall TAT
Amplification				
1. Add PCR mixture to PCR tube	00:02:34	00:04:46	02:22:48	02:27:34
2. Add extracted DNA to PCR tube	00:02:10			
3. Run in PCR system	02:22:48			
Gel electrophoresis				
4. Add PCR product to agarose gel	00:05:43	00:07:49	00:27:19	00:35:08
5. Electrophoresis	00:27:19			
6. Interpret the bands with an analyzer	00:02:06			
PCR product clean-up				
7. Add clean solution and run in the PCR system	00:24:30	00:02:30	00:22:00	00:24:30
Sequencing reaction				
8. Dilute the PCR products	00:03:02	00:03:02	02:33:57	02:36:59
9. Add primer and dye mixture to plate				
10. Add diluted PCR products to plate				
11. Run in PCR system	02:33:57			
Purification for dye removal				
12. Add sodium acetate/EDTA buffer	00:01:36	00:19:32	05:56:57	06:16:29
13. Add ethanol to plate	00:01:36			
14. Centrifuge (30 min at 2000× *g*)	00:31:10			
15. Invert to a paper towel and remove supernatant	00:00:47			
16. Add 80% ethanol	00:01:35			
17. Centrifuge (5 min at 2000× *g*)	00:05:49			
18. Invert to a paper towel and remove supernatant	00:00:49			
19. Dry 3 min at 65 °C	00:03:15			
20. Add high-deionized formamide	00:02:41			
21. Dry 3 min at 95 °C	00:03:12			
22. Incubate for 3 min at 0 °C	00:03:12			
23. Run in genetic analyzer	05:20:47			
Interpretation and report				
24. Interpret the results	00:11:28	00:10:00	00:01:28	00:11:28
25. Input the result on LIS manually				
Total	12:32:06	00:47:39	11:44:29	12:32:06

Abbreviations: TAT, turn-around time; SBT, sequence-based typing; HLA, human leukocyte antigen; PCR, polymerase chain reaction; DNA, deoxyribonucleic acid; EDTA, ethylenediamine tetraacetic acid; LIS, laboratory information system.

**Table 5 diagnostics-14-01793-t005:** TAT assessment in NGS for HLA typing.

Processing Step	TAT	Hands-On Time	Machine Running Time	Overall TAT
Amplification				
1. Add nuclease-free water to PCR tube	00:01:21	00:14:12	04:40:00	04:54:12
2. Add polymerase, buffer, and dNTPs	00:04:08			
3. Identify and add primer pellet	00:08:43			
4. Add extracted DNA to reaction mix	00:09:19			
5. Run in PCR system	04:40:00			
DNA quantification				
6. Add Qubit reagent to Qubit tube	00:20:22	00:47:46	00:00:00	00:47:46
7. Add Qubit buffer to Qubit tube				
8. Add DNA to Qubit tube	00:10:23			
9. Measure amplicons conc. at fluorometer	00:06:35			
10. Calculate the required volume of each gene and pool amplicons	00:10:26			
Fragmentation and adapter ligation				
11. Prepare fragmentation master mix	00:08:23	00:14:16	00:45:45	01:00:01
12. Add master mix to plate well				
13. Add amplicon to plate well	00:01:32			
14. Incubation in a PCR system at 25 °C for 20 min and 70 °C for 10 min	00:30:45			
15. Prepare adaptor ligation master mix	00:04:21			
16. Add ligation master mix to DNA fragmentation				
17. Incubation in PCR system at 20 °C for 15 min	00:15:00			
DNA clean-up				
18. Add magnetic beads and incubate for 5 min	00:07:52	00:27:35	00:00:00	00:27:35
19. Place the plate on a magnetic stand and remove supernatant	00:05:26			
20. Add 80% ethanol, incubate for 30 s, and remove the supernatant. Repeat 3 times	00:04:13			
21. Remove ethanol and dry 3~5 min	00:01:37			
22. Add elution buffer and incubate for 2 min at RT on shaker	00:03:12			
23. Place the plate on a magnetic stand and transfer eluate to a new plate	00:05:15			
Indexing PCR				
24. Prepare (thaw and centrifuge) for reagent	00:06:33	00:09:30	00:22:00	00:31:30
25. Add HiFi PCR Mix to IndX plate	00:01:13			
26. Add eluate to IndX plate	00:01:44			
27. Run in PCR system	00:22:00			
Library pooling, DNA clean-up, and size selection				
28. Transfer DNA to new tube and mix	00:01:52	00:22:30	00:00:00	00:22:30
29. Add magnetic beads and incubate for 5 min	00:05:41			
30. Place the plate on a magnetic stand and remove the supernatant	00:05:26			
31. Add 80% ethanol, incubate for 30 s, and remove supernatant	00:01:26			
32. Remove ethanol and dry 3~5 min	00:03:09			
33. Add elution buffer and incubate for 2 min at RT on shaker	00:02:27			
34. Place the plate on a magnetic stand and transfer eluate to a new tube	00:02:29			
Library quantification				
35. Prepare Qubit working solution with library sample	00:06:09	00:09:04	00:00:00	00:09:04
36. Measure library conc. at a fluorometer and calculate Qubit results to nanomolar conc.	00:02:55			
Next-generation sequencing				
37. Prepare (thaw and wash) for reagent	00:07:13	00:13:43	17:00:00	17:13:43
38. Dilute library to a conc. of 4 nM				
39. Add 0.2N NaOH				
40. Incubation the tube	00:05:00			
41. Add HT1 buffer	00:01:30			
42. Run paired-end sequencing in Miseq analyzer	17:00:00			
Interpretation and report				
43. Interpret the results	01:01:40	01:01:40	00:00:00	01:01:40
44. Input the result on LIS manually				
Total	26:47:20	03:59:35	22:47:45	26:47:20

Abbreviations: TAT, turn-around time; NGS, next-generation sequencing; HLA, human leukocyte antigen; PCR, polymerase chain reaction; dNTPs, deoxynucleotide triphosphates; conc., concentration; RT, room temperature; HT, hybridization; LIS, laboratory information system.

**Table 6 diagnostics-14-01793-t006:** Comparison of risk and TAT between SBT and NGS.

	SBT	NGS	*p* Value
Risk			
Total steps, N	25	44	NA
Acceptable steps, N (%)	25 (100)	44 (100)	NA
Unacceptable steps, N (%)	0 (0)	0 (0)	NA
Total RPN	465	1169	NA
* RPN, median (range)	12 (3–72)	23 (12–62)	0.001
Severity, median (range)	3 (2–8)	5 (3–8)	0.005
Occurrence, median (range)	1 (1–2)	1 (1–2)	0.446
Detectability, median (range)	4 (1–6)	5 (3–6)	<0.001
TAT			
Total TAT, hh:mm:ss	12:32:06	26:47:20	NA
Total hands-on time, hh:mm:ss	00:47:39	03:59:35	NA
Hands on time/step, hh:mm:ss	00:01:54	00:05:13	NA

* The median values of RPN, severity, occurrence, and detectability for SBT and NGS were compared using a Kruskal–Wallis test. The severity, occurrence, and detectability scores were calculated using FMEA, and acceptable and unacceptable steps were accessed using the RAM rating scale. Abbreviations: SBT, sequence-based typing; NGS, next-generation sequencing; RPN, risk priority number; TAT, turn-around time; NA, not available; RAM, risk acceptability matrix; FMEA, failure mode and effect analysis.

## Data Availability

All data analyzed during this study are included in this published article and the datasets generated during this study are available from the corresponding author on reasonable request.

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
