# Peer review of "Comparative Assessment of Risk and Turn-Around Time between Sequence-Based Typing and Next-Generation Sequencing for HLA Typing"

_diagnostics, 2024, doi:10.3390/diagnostics14161793_

Round 1

Reviewer 1 Report

Comments and Suggestions for Authors

In this manuscript entitled “Comparative Assessment for Risk and Turn-around Time Between Sequence-based Typing and Next-generation Sequencing for HLA Typing”, the authors evaluated the relative risk and turn-around time (TAT) of sequence-based typing (SBT) and next-generation sequencing (NGS) in human leukocyte antigen (HLA) typing. They sought to address the efficiency and reliability of these methods using failure mode and effects analysis (FMEA) and a risk acceptability matrix (RAM). The authors found that NGS analysis revealed a higher total risk priority number (RPN) score than SBT. The mean RPN, severity, and detectability scores were also higher for NGS, indicating a higher risk level. NGS also exhibited longer total TAT. The authors described that the increased manual steps and techniques required for the NGS analysis contributed to its higher risk and longer TAT. They suggested that automating certain NGS processes improves efficiency. This study provides valuable information regarding the laboratory efficiency of HLA typing methods through the evaluation of FMEA and RAM. While NGS offers advanced capabilities, the authors explained that its current implementation involves higher risks and longer processing times than SBT. Following are my specific concerns.

Major comments

1.    Many key citations are missing and the authors should add essential citations including the following ones. On page 1, please cite the papers describing the HLA typing (Edgerly et al., PMID: 29858798, Mayor et al., PMID: 26018555). Add citations in the sentence describing the transplantation and autoimmune diseases (Yoo et al., PMID: 37968983, Kim et al, PMID: 35950451). On page 2, please cite the key papers regarding the application of NGS in the laboratory (Yin et al., PMID: 33892986 Jung and Lee, PMID: 36859476).

2.    In the results, the authors evaluated the relative risk of SBT compared to NGS using FMEA and RAM. I wonder why the authors chose FMEA and RAM as evaluation tools. Please explain in detail the reasons for selecting these tools, considering their features. Furthermore, the authors need to describe the significance and meaning of the data in the Result section to highlight the finding of this study. Additionally, the authors should explain the significance of the difference in range scores between SBT and NGS.

3.    The authors showed the potential defects in all processing steps of SBT and NGS. I wonder what criteria were used for identifying a potential defect. I think that the contamination issue can arise in steps other than 11, 23, 24, and 28 in the NGS processing steps. The authors should provide clear criteria for identifying potential defects.

4.    The authors compared the RPN scores between SBT and NGS (456 vs. 1,171). However, the total RPN score for NGS is 1,169, not 1,171. Additionally, the authors compared the total TAT between SBT and NGS (12:32:08 vs. 26:47:20). The total TAT for SBT is 12:32:06, not 12:32:08. The authors need to correct these inaccuracies and double-check for any other errors in score notations throughout the paper.

Minor comments

1.    In the introduction, there are some sentences that are missing references, such as sentence 38. Please include the appropriate references.

2.    In the table 3 and 5, please provide detailed information about the experimental conditions, such as incubation temperature.

3.    The authors need to provide the full form of the HT1 buffer abbreviation in table legend.

4.    In table 6, the authors should clarify which test the p-value was derived from in legend.

5.    Please correct minor typos throughout the manuscript. Following are some examples.

-       On page 1, line 11, please correct “Background” to “Background: ”.

-       On page 3, line 99, please remove “6”.

-       On page 10, line 244, please correct “16” to “[16]”.

Comments on the Quality of English Language

Overall fine

Author Response

Reviewer #1:

In this manuscript entitled “Comparative Assessment for Risk and Turn-around Time Between Sequence-based Typing and Next-generation Sequencing for HLA Typing”, the authors evaluated the relative risk and turn-around time (TAT) of sequence-based typing (SBT) and next-generation sequencing (NGS) in human leukocyte antigen (HLA) typing. They sought to address the efficiency and reliability of these methods using failure mode and effects analysis (FMEA) and a risk acceptability matrix (RAM). The authors found that NGS analysis revealed a higher total risk priority number (RPN) score than SBT. The mean RPN, severity, and detectability scores were also higher for NGS, indicating a higher risk level. NGS also exhibited longer total TAT. The authors described that the increased manual steps and techniques required for the NGS analysis contributed to its higher risk and longer TAT. They suggested that automating certain NGS processes improves efficiency. This study provides valuable information regarding the laboratory efficiency of HLA typing methods through the evaluation of FMEA and RAM. While NGS offers advanced capabilities, the authors explained that its current implementation involves higher risks and longer processing times than SBT. Following are my specific concerns.

Major comment

1) Many key citations are missing and the authors should add essential citations including the following ones. On page 1, please cite the papers describing the HLA typing (Edgerly et al., PMID: 29858798, Mayor et al., PMID: 26018555). Add citations in the sentence describing the transplantation and autoimmune diseases (Yoo et al., PMID: 37968983, Kim et al, PMID: 35950451). On page 2, please cite the key papers regarding the application of NGS in the laboratory (Yin et al., PMID: 33892986 Jung and Lee, PMID: 36859476)

Thank you for your thorough review and valuable comment on this manuscript. We have carefully considered your suggestions regarding the inclusion of key citations and have made the necessary revisions to the manuscript accordingly. We believe these additions have significantly strengthened the manuscript and provided the necessary context for our findings.

Specifically:

On page 1, we have added the recommended citations for HLA typing, referencing the papers by Edgerly et al. (PMID: 29858798) and Mayor et al. (PMID: 26018555).

In the section describing transplantation and autoimmune diseases, we have now cited the relevant works by Yoo et al. (PMID: 37968983) and Kim et al. (PMID: 35950451).

On page 2, we have included the key papers on the application of NGS in the laboratory as suggested, citing Yin et al. (PMID: 33892986) and Jung and Lee (PMID: 36859476).

Human leukocyte antigen (HLA) typing plays a pivotal role in determining compatibility between donor and recipients in transplantation and providing insights into genetic basis of autoimmune diseases [1-4]. (Page 1, line 29)

Next-generation sequencing (NGS) has recently been actively introduced for HLA typing, utilizing massive parallel sequencing of clonal DNA molecules [8,9]. (Page 1, line 41)

We added/modifired the references.

  1. Edgerly, C.H.; Weimer, E.T. The Past, Present, and Future of HLA Typing in Transplantation. Methods Mol Biol. 2018, 1802, 1-10.
  2. Mayor, N.P.; Robinson, J.; McWhinnie, A.J.M.; Ranade, S.; Eng, K.; Midwinter, W.; Bultitude, W.P.; Chin, C.H.; Bowman, B.; Marks, P.; Braund, H.; Madrigal, J.A.; Latham, K.; Marsh, S.G.E. HLA Typing for the Next Generation. PLoS One. 2015, 10, e0127153.
  3. Yoo, H.J.; Yi, Y.; Kang, Y.; Kim, S.J.; Yoon, Y.I.; Tran, P.H.; Kang, T.; Kim, M.K.; Han, J.; Tak, E.; Ahn, C.S.; Song, G.W.; Park, G.C.; Lee, S.G.; Kim, J.J.; Jung, D.H.; Hwang, S.; Kim, N. Reduced Ceramides Are Associated with Acute Rejection in Liver Transplant Patients and Skin Graft and Hepatocyte Transplant Mice, Reducing Tolerogenic Dendritic Cells. Mol Cells. 2023, 46, 688-699.
  4. Kim, G.R.; Choi, J.M. Current Understanding of Cytotoxic T Lymphocyte Antigen-4 (CTLA-4) Signaling in T-Cell Biology and Disease Therapy. Mol Cells. 2022, 45, 513-521.
  5. Yin, Y.; Butler, C.; Zhang, Q. Challenges in the application of NGS in the clinical laboratory. Hum Immunol. 2021, 82, 812-819.
  6. Jung, S.; Lee, J.S. Single-Cell Genomics for Investigating Pathogenesis of Inflammatory Diseases. Mol Cells. 2023, 46, 120-129.

2) In the results, the authors evaluated the relative risk of SBT compared to NGS using FMEA and RAM. I wonder why the authors chose FMEA and RAM as evaluation tools. Please explain in detail the reasons for selecting these tools, considering their features. Furthermore, the authors need to describe the significance and meaning of the data in the Result section to highlight the finding of this study. Additionally, the authors should explain the significance of the difference in range scores between SBT and NGS.

Thank you for insightful comment and for giving us the opportunity to clarify our methodology. The CLSI guideline (EP18-A3) recommends several risk management techniques, including FMEA, FTA, and FRACAS, to identify and control potential failure sources in the laboratory setting. We selected FMEA and RAM for our study based on the following consideration. FMEA utilizes a bottom-up approach, allowing us to systematically identify and evaluate potential issues in each testing step or component. FTA, on the other hand, employs a top-down approach, focusing on identifying the most critical failures and tracing their possible causes. We determined that FMEA’s capability to address potential failures across the entire process made it more suitable for our study. FRACAS is distinct from FMEA and FTA, as it records actual failures and applies corrective actions based on past events rather than hypothesizing potential issues. Given that our study aimed to preemptively identify and mitigate risks, FMEA was deemed the most appropriate method. Furthermore, we considered it meaningful to compare the results from FMEA with those obtained from RAM, as suggested by the CLSI guidelines for comprehensive risk evaluation. Furthermore, we have revised the Results section to provide a more detailed explanation of the significance and meaning the data. Lastly, we have revised the data presentation in Table 6 to show the range values and have added a corresponding explanation in the Results section to clarify this change.

FMEA systematically identifies and evaluates potential failures and their causes across the entire process by a top-down approach [17]. (Page 2, line 54)

All steps revealed varying levels of risk, but according to RAM rating scale, the risks were clinically acceptable. (Page 4, line 143), (Page 5, line 153)

The risks at all steps in both SBT and NGS were clinically acceptable. (Page 9, line 172)

3) The authors showed the potential defects in all processing steps of SBT and NGS. I wonder what criteria were used for identifying a potential defect. I think that the contamination issue can arise in steps other than 11, 23, 24, and 28 in the NGS processing steps. The authors should provide clear criteria for identifying potential defects

Thank you for your valuable comment. The process of identifying potential defects involved each team members conducting individual evaluations based on literature review and personal experiences, followed by a collective reassessment as a team. Contamination is indeed a critical issue in PCR-based techniques such as SBT and NGS, and it is a significant cause of failure in many steps. However, due to word limitation in the paper, we condensed the content to focus on 1-2 potential defects deemed most important at each step to convey specific points effectively. I hope you understand and accept these limitations.

4) The authors compared the RPN scores between SBT and NGS (456 vs. 1,171). However, the total RPN score for NGS is 1,169, not 1,171. Additionally, the authors compared the total TAT between SBT and NGS (12:32:08 vs. 26:47:20). The total TAT for SBT is 12:32:06, not 12:32:08. The authors need to correct these inaccuracies and double-check for any other errors in score notations throughout the paper.

Thank you for pointing out these inaccuracies in our manuscript. We have corrected the total RPN score for NGS to 1,169 and the total TAT for SBT to 12:32:06. Additionally, we have thoroughly reviewed the manuscript the ensure that all score notations and data are accurate and consistent throughout the paper.

Minor comments

  • In the introduction, there are some sentences that are missing references, such as sentence 38. Please include the appropriate references.

Thank you for your comment. We have added the appropriate references to the sentences in the Introduction, including sentence 38, where reference was previously missing. We believe these additions provide the necessary context and support for our statements.

Sequence-based typing (SBT) is considered the gold standard at the allele level for determining the full nucleotide sequence of the alleles present [7]. (Page 1, line 38)

Clinical laboratories are consistently under pressure to provide qualified and reliable results within a limited time, without compromising laboratory efficiency [11]. (Page 2, line 46)

  • In the table 3 and 5, please provide detailed information about the experimental conditions, such as incubation temperature.

Thank you for your thoughtful comment. Due to space constraints, we concentrated on addressing potential failures at each step rather than detailing the specific conditions of the protocol. Nevertheless, in response to your valuable feedback, we have made every effort to include as many relevant experimental conditions as space allows.

  • The authors need to provide the full form of the HT1 buffer abbreviation in table legend.

Thank you for your comment. Following an inquiry with the manufacturer, we confirmed that “HT1 buffer” is not an abbreviation and actually refers to the hybridization buffer. We have revised the manuscript to reflect this clarification.

Abbreviations: NGS, next generation sequencing; HLA, human leukocyte antigen; FMEA, failure mode and effect analysis; RAM, risk acceptability matrix; S, severity; O, occurrence; D, detectability; RPN, risk priority number; PCR, polymerase chain reaction; IA, incorrect amounts; MA, missed addition; R, repeat; D, delay; IR, incorrect result; N, negligible; I, improbable; dNTPs, deoxynucleotide triphosphates; Rm, Remote; M, minor; temp., temperature; conc., concentration; Oc, occasion; RT, room temperature; HT, hybridization; LIS, laboratory information system. (Page 6, Table 3)

Abbreviations: TAT, turn-around time; NGS, next generation sequencing; HLA, human leukocyte antigen; PCR, polymerase chain reaction; dNTPs, deoxynucleotide triphosphates; conc., concentration; RT, room temperature; HT, hybridization; LIS, laboratory information system. (Page 8, Table 5)

HT, hybridization

  • In table 6, the authors should clarify which test the p-value was derived from in legend.

Thank you for your comment. We have clarified in the table legend that the p-value in Table 6 were derived from a Kruskal-Walllis-test comparing the mean values of RPN, severity, occurrence, and detectability for SBT and NGS.

The mean values of RPN, severity, occurrence, and detectablility for SBT and NGS were compared using a Kruskal-Wallis test. (Page 9, Table 6)

  • Please correct minor typos throughout the manuscript. Following are some examples.

-       On page 1, line 11, please correct “Background” to “Background: ”.

-       On page 3, line 99, please remove “6”.

-       On page 10, line 244, please correct “16” to “[16]”.

Thank you for your comment. According to your comment, we corrected all typo and double-checked.

This response acknowledges the reviewer's suggestions and addresses the reviewer’s request by indicating that the necessary revisions have been made to improve the clarity and significance of the study’s findings.

Reviewer 2 Report

Comments and Suggestions for Authors

If possible increase the sample size. Can you also include cost-benefit analysis study between the two methods. 

Author Response

Reviewer #2:

If possible increase the sample size. Can you also include cost-benefit analysis study between the two methods.

Thank you for your important comment. Regarding your suggestion to increase the sample size, our analyses indicated that, under typical conditions, there was minimal difference in turnaround time (TAT) between measurements with a single sample and those with larger sample sizes. Consequently, we proceeded with the current sample size for our analysis.

Additionally, we conducted a cost-benefit analysis; however, due to the significant cost associated flow cells in NGS and the considerable variation depending on sample size and test parameters, the analyses faced inherent limitations. As a result, we decided to omit this section from the paper. We appreciate your understanding and consideration of these aspects.

This response acknowledges the reviewer's suggestions, explains the rationale behind the decisions made, and shows appreciation for the reviewer’s understanding.
